# Sustainable or Not? Insights on the Consumption of Animal Products in Poland

**DOI:** 10.3390/ijerph192013072

**Published:** 2022-10-11

**Authors:** Katarzyna Mazur-Włodarczyk, Agnieszka Gruszecka-Kosowska

**Affiliations:** 1Faculty of Economics and Management, Opole University of Technology, 7 Luboszycka St., 45-036 Opole, Poland; 2Department of Environmental Protection, Faculty of Geology, Geophysics, and Environmental Protection, AGH University of Science and Technology, al. A. Mickiewicza 30, 30-059 Krakow, Poland

**Keywords:** meat products, fish products, egg products, consumption habits, sustainable consumption, harmonization

## Abstract

Animal products are one of the main constituents of the human diet. They are the main source of energy, proteins, microelements, and bioactive substances. The most popular negative health impacts linked with the consumption of animal products are obesity, atherosclerosis, heart attacks, and cancer. Apart from human health, consuming animal products is also controversial lately, due to farm animals’ well-being and environmental protection issues. Thus, within the context of sustainability, the consumption trends of animal products were investigated through our on-line questionnaire surveys. The following animal products were involved in the survey: unprocessed meat (pork, beef, lamb, veal, mutton, chicken, duck, goose, turkey), processed meat (cold-cuts, sausages, pates), fish products, and eggs. Our research concluded that consumption among respondents with higher education was unsustainable for both unprocessed and processed meat, as eating habits in terms of type and quantity of consumed meat indicated respondents’ unawareness. The consumption of fish products was also revealed as unsustainable regarding the quantity of fish consumed in terms of its beneficial nutritional values. Egg consumption was revealed as the most sustainable among the investigated animal products. However, insignificant egg consumption among the respondents showed the actual need of social education in terms of the current knowledge regarding the beneficial aspects of eggs.

## 1. Introduction

In the 21st century, the need to care for the natural environment and the future of next generations have become an essential part of the research in most scientific disciplines. According to the latest strategy for sustainable development, also known as the 2030 Agenda [1], signed in 2015, harmonious development is potentially achievable through the implementation of seventeen Sustainable Development Goals (SDGs). 

The twelfth of the Sustainable Development Goals, called Responsible Consumption and Production, aims to ensure fundamental changes for moving towards more sustainable patterns of consumption and production by the year 2030 [1]. In this context, the terms are used interchangeably [2]. The first narrower concept is sustainable consumption, which means consumption in such a manner that allows for the preserving of resources and the environment as much as possible for future generations. The second more wider concept is responsible consumption which describes taking under consideration environmental, economic, and social aspects during the consumption. There are two main types of actions that may reduce the negative environmental impact of consumption. The first group includes controlling production and limiting consumption as they both impact the natural environment [3]. The second type of activities has been designed to develop specific consumption habits, such as rejecting products that are unsustainable, replacing harmful products with less or harmless ones, or reducing the consumption of some products in order to make them available for future generations. However, much of this depends on the change of consumers’ habits. In this case, moral considerations prevail over social dilemmas [4], therefore, it is essential to become individually interested and consciously implement sustainable changes. Another issue related to the above-mentioned topic is conscious eating. This term denotes the consumption of food in accordance with the principles of healthy eating, as well as making conscious product choices [5]. Sustainable consumption is associated with many dilemmas [4] and new trends [5], as well as changes in guidelines for healthy and sustainable eating [6]. The need to focus on sustainable consumption of products intended for humans is visible in research in many countries [7,8,9,10]. The purpose of their conduct is lofty, but it is worth noting that these studies are not standardized. Various researchers consider the aspect of sustainable consumption of food products through narrow sections of it. Sustainable consumption of animal products is most often analyzed into subgroups, including: meat [8,11,12,13], fishes [14,15,16,17], and eggs [18,19,20,21], in which frequently declared consumer attitudes are analyzed [22]. Based on the above, our research has tried to associate consumption reports with specific questions about lifestyle and choice preferences to contribute to global sustainability through the European context.

In terms of animal food products, the nutritional recommendations of more advanced countries, such as Germany, Brazil, Sweden, and Qatar, suggest that one should lean towards [6]: a plant-based diet, consumption of white meat (rather than the meat of large animals for slaughter and wild game animals), buying unprocessed meat, maintaining a diet rich in fish products, eating products with a lower fat content, choosing organic/ecological products. In terms of more environmentally friendly consumption, it is also important to choose food products with a relatively lower carbon footprint. The footprint refers both to carbon dioxide emissions and the excessive use of raw materials, such as water, energy, protective measures, and fertilizers. The following mitigation actions would be suggested [12,22,23,24]: purchasing more plant products since intensive animal husbandry (meat, fish, seafood) results in the overproduction of greenhouse gases: purchasing local products and minimizing the environmental costs of transporting goods; purchasing more unprocessed products, which would allow for the minimization of the environmental costs of producing processed food; limiting the waste of food which has been already purchased by the optimal management of the food supply; sourcing and distributing food more rationally and avoiding food waste, for example, by paying more attention to expiry dates, preparing portions of adequate size, and proper storage. An extreme movement of freeganism is also worth noting as it focuses on the consumption of ‘wasted’ food, i.e., food that is intended to be thrown away (e.g., food slightly spoiled or past its use-by date) [25].

In recent years, a new meat consumption tendency has become evident among Poles. The majority of them declare they would like to eat less meat, but not necessarily become vegans or vegetarians [26]. Since traditional Polish cuisine is based on an old-fashioned pork chop, the willingness of Poles to reduce the amount of meat consumed weekly shows the beginning of the transformation. Regarding the national consumption, in 2019 Poles consumed 61 kg per person (5.08 kg per month per person), and reduced their meat intake by 8.8%, in comparison with 2010 (5.57 kg per month per person) [27] (p. 336). Among the reasons of eliminating meat from their plate in 2021 [28] beside the most popular health issues (53%) and improving their own well-being (42%), the following reasons were also stated in explaining the decreasing trend: limiting animal suffering (31%), distrust of farmer products (31%), concern for the environment (30%), the taste of plant-based foods (26%), changes in the diet of loved ones (19%), financial aspects (10%). Nevertheless, still the amount of meat consumed exceeds the goals of a sustainable diet [13].

Unfortunately, the existing scientific research does not comprehensively discuss the consumption of all types and amounts of animal products individually. However, this information becomes essential in two main research areas. The first is related to the science of nutrition, which outlines new nutritional trends to improve the functioning of the body and avoid the diseases related to civilization, e.g., obesity [29], atherosclerosis [30], and cancers [31]. The second research area is related to the fact that our food—along the various stages of the food chain—may be contaminated when reaching a dining table. Some of these factors can be easily eliminated by choosing the type and quality food, maintaining kitchen hygienic standards when preparing animal products, and the temperature of food processing [32,33]. Unfortunately, elimination of other determinants requires systemic actions as they refer to the environmental pollution of food and feed [34,35,36]. Regarding the above, in the context of risk related with food consumption and its prevention strategies the major role plays the consumption rate among related subpopulations (e.g., children, seniors, manual workers, etc.) among exposed humans. 

Creation of the trend in consumption is the long process requiring mental changes in the society, which does not only rely on style and fashion. Taking this into consideration, the aim of this study was to determine the consumption preferences regarding various types and amounts of animal products consumed jointly, belonging to the most common animal products eaten in Poland, i.e., meat (processed and not processed), fish, and eggs. Based on the collected data, the authors discussed the consumption habits of Poles in terms of the guidelines for sustainable development. Our research has also contributed to promoting the issues of environmentally sustainable consumption of animal products.

## 2. Materials and Methods

In order to determine the food preferences in animal products, the authors designed a questionnaire study. The survey was conducted in 2017 (between February and November) using the Interankiety.pl platform (in a digital form). The survey was conducted using a non-probability, exponential, non-discriminative snowball sampling (QuestionPro [37]) where existing respondents recruited further subjects from among their acquaintances. The sampling was virtual, as the survey was prepared in digital form and mainly scientific (like ResearchGate or LinkedIn) and social (like Meta) networks were used for the dissemination of the link to the survey. 

The conducted survey discussed the following groups of animal products popular in Poland: unprocessed meat used for human consumption: beef, veal, pork, lamb, lamb, poultry (chickens and roosters, turkeys, ducks, geese), processed meat: cold cuts, sausages, pates, canned meat, sea and freshwater fish, fresh, chilled, frozen, dried, smoked, or salted food, and eggs of domestic fowl.

The questionnaire consisted of 39 questions divided into three parts. The first part was a general section, in which the respondents were asked 5 questions regarding where they buy animal products, in particular: unprocessed meat, processed meat, fish (fresh or frozen), and eggs. In this part of the survey, respondents were asked to choose the most suitable answers from the given propositions. For unprocessed meat products, the possible answers for the respondents to choose were as follows: “market”, “supermarket”, “meat shop”, “at a butcher’s”, “butcher shop”, “neighborhood shop”, “health-food store”, “own production”, “not applicable”, “refusal to answer”. For processed meat, possible answers were the following: “market”, “supermarket”, “sausage shop”, “butcher shop”, “neighborhood shop”, “health-food store”, “own production”, “not applicable”, and “refusal to answer”. For fish products, the options of answers were the following: “market”, “supermarket”, “fish shop”, “neighborhood shop”, “health-food store”, “own fishing”, “not applicable”, and “refusal to answer”. For eggs, the answers to choose from were as follows: “market”, “supermarket”, “neighborhood shop”, “health-food store”, “from farmer”, “not applicable”, and “refusal to answer”. Regarding egg consumption, an additional question related with preference of buying eggs regarding hen raising methods (specifically about marking the code number) was asked. In this question, the answers to choose from were as follows: “0—organic egg production”, “1—free range eggs”, “2—deep litter indoor housing”, “3—cage farming”, “I do not know”, and “refusal to answer”. Additionally, in the case of all investigated animal products, if the respondents did not find the relevant answer, they could choose the option “other” and provide their own answer.

In the second part of the questionnaire, 25 questions were asked about the frequency and quantity of the one-time consumption of the investigated animal products commonly consumed in Poland. Participants were asked to estimate how much food they consume in a serving. To do so, two different options were proposed. One more specific (in grams) and the other broader one (portion size), where relevant. In this section, respondents were asked to choose the most suitable answers from the given propositions. According to the frequency of the consumption of animal products, the answers were stated as follows: “more than three times a day”, “three times a day”, “twice a day”, “once a day”, “six times a week”, “five times a week”, “four times a week”, “three times a week”, “twice a week”, “once a week”, “several times a month”, “dozen times a year”, “several times a year”, “I do not eat it at all”. According to the weight of the portion of the consumed animal products, the possible answers were as follows: “50 g”, “100 g”, “200 g”, “300 g”, “400 g”, “500 g”, “750 g”, “1 kg”. In the case of the processed meat products, visual units were also added for those respondents who preferred a visual description. They were as follows: “1 slice”, “2 slices”, “3 slices”, “4 slices”. In the case of eggs consumed at one serving, the possible answers were as follows (number of eggs): “1/2”, “1”, “2”, “3”, “4”, “5”, “6”, “7”. Again, if the respondents did not find the relevant answer, they could choose the option “other” and provide their own answer.

Sociodemographic questions were asked in the third part of the questionnaire. Respondents were asked about their gender, age, educational level, marital status, region (voivodeship/province), area of residence regarding the number of inhabitants, number of people in the household, and indicative net income. Details regarding the possible answers in particular questions of the sociodemographic part of the survey are given together with the results in Table 1.

For further investigations, only complete questionnaires were processed for subsequent investigation. This means that only questionnaires that provided answers to all questions in all three parts of the survey (“other” or “refuse to answer” were considered as a given answer) were valid. As our survey was detailed and thus might be seen as tedious by the respondents ultimately, we have collected 67 complete questionnaires, in which all questions were answered. The questionnaires were also completed by adults coming from and living in Poland. Respondents also declared consuming animal products and acting as the main person supplying their households with food products. 

The research approached this topic from a qualitative perspective. The main aim of the studies was to receive the general trend of joint consumption of various animal products, therefore, the multiple answers were eligible in our survey on consumption of meat and fish food products. However, this approach prevented the possibility of performing the statistical analysis due to the variation of the total n number of respondents as multiple answers were given to the majority of the questionnaire questions. Thus, the results of our study were only analyzed in the descriptive manner. Moreover, in accordance with the above the following research questions (RQ) were formulated in the research process: 

RQ1: Was the consumption of animal products in Poland sustainable in accordance with the modern trend of choosing a healthy lifestyle-special care for a balanced, sustainable diet?

RQ2: Did the aspect of sustainable consumption of animal products also apply to purchasing meat, fish, and eggs in health food stores?
ijerph-19-13072-t001_Table 1Table 1Sociodemographic characteristic of Polish respondents.
Demographic FactorFrequency(*n* = 67)Percentage (%)GenderMale1624Female5176Prefer not to answer00Age18–20 years0021–30 years142131–40 years233441–50 years101551–60 years6961–70 years69Over 70 years57Refusal to answer33Educational levelSecondary education46Secondary vocational23Post-secondary69Higher vocational35Bachelor degree46Master degree4467Refusal to answer44Marital statusSingle2131Married/in relation3857Separation/after divorce23Widowed46Refusal to answer23Single2131Region of Poland(voivodeship/province)Dolnośląskie1015Kujawsko-pomorskie11Łódzkie23Małopolskie2436Mazowieckie34Opolskie1725Podkarpackie23Śląskie23Wielkopolskie34Refusal to answer34Area of residence,number of inhabitantsCountryside agricultural area913City, up to 20,00011City, 21,000–100,00046City, 101,000–250,0001624City, 251,000–500,00034City, 501,000–750,00046City, 751,000–1,000,0001827City, over 1,000,000913Refusal to answer34Number of people in the household11015227403101541116558611711Refusal to answer23Indicative net income in PLN [in USD]Up to PLN1000 [US$251.5]34PLN1001–3000 [US$251.7–754.5]3045PLN3001–5000 [US$754.8–1257.6]1116PLN5001–7000 [US$1257.8–1760.6]710PLN7001–9000 [US$1760.8–2263.6]00Over PLN9000 [US$2263.6]34Refusal to answer1319


However, single choice answers were obtained in the part of questionnaire related with marking of the eggs (class 0, 1, 2, or 3) bought by the respondents. Thus, the statistical analysis of the Chi-squared test was performed using a Microsoft Excel 2007 spreadsheet in order to test the following research hypotheses related with dependence of egg purchasing with sociodemographic features: gender, marital status, level of education, and income: 

**H0.** 
*The purchased egg marking did not depend on sociodemographic characteristics,*


**H1.** 
*The purchased egg marking depended on socio-demographic characteristics.*


In the case when calculated *p*-values were >0.05 it confirmed the validity of the null hypothesis (H0), while *p*-values <0.05 confirmed the validity of the alternative hypothesis (H1).

## 3. Results

### 3.1. The Characteristic of the Respondents

In our results, we obtained only 67 fully completed questionnaires. Based on the results received we could state that majority of the respondents were women (76%). Most often, the participants were aged between 20 and 50 years old (70%). 67% of the respondents graduated from a university and had received their master’s degrees. Participants, who supplied households with animal products were mainly married, in an informal relationship (57%) or single (31%). The dominant group of respondents declared living in the following provinces: Małopolskie (36% of respondents), Opolskie (25%) and Dolnośląskie (15%) regions. The majority of the respondents declared living in cities (84%). Most of them lived in large cities: 27% for cities between 751,000 and 1,000,000 inhabitants, 24% between 101,000 and 750.000 inhabitants, and 13% for cities above 1,000,000 inhabitants. Living in rural areas was declared by the 13% of the respondents. The respondents’ households most often consisted of two (40%), four (16%), and three people or one person (15% each). On the other hand, the net income per person in a household usually ranged between PLN1001–3000 (US$238–712) and PLN3001–5000 (US$713–1188). Detailed sociodemographic characteristics are presented in Table 1. The fact that the majority of our respondents possessed higher education that was also related with the income rate could have an impact on the answers given to questions on sustainability of the animal products consumption in Poland. Based on that in our further description of the results we will underline the fact that majority of answers were given by highly educated respondents.

### 3.2. The Characteristic of the Supply Sources of Animal Products

Most of the respondents in our studies, who appeared to be highly educated people, bought animal products in supermarkets: unprocessed meat (57% of the answers), processed meat (51%), fish (60%), eggs (42%); at the butcher’s: unprocessed meat (60%), processed meat (69%); and in a local store: unprocessed meat (37%), processed meat (39%) and eggs (43%). 61% of the respondents declared that they buy fish products in fish stores. In health food stores, respondents declared buying eggs (15%), processed meat (6%), unprocessed meat (3%), and fish (1%). It should be emphasized that the share of organic food in the basket of products is lower than in the case of non-organic food, mainly due to the price and availability of certified food [38]. 25% of the respondents bought eggs at the market. Detailed results are presented in Figure 1.

When supplying their households with eggs, the respondents chose only hen eggs. 16% of the participants generally did not pay attention to the egg marking allowing consumers to distinguish free range eggs and organic farming eggs from the industrial caged hen production (Figure 2). The largest number of respondents bought eggs marked as 1—free range eggs, (34%), then 0—organic egg production (18%), 2—deep litter indoor housing (12%), and finally 3—cage farming (6%). There were no significant differences in the responses of the respondents representing different socio-economic groups.
Figure 1Supply sources of unprocessed and processed meat, fish, and eggs [%] regarding to respondents.
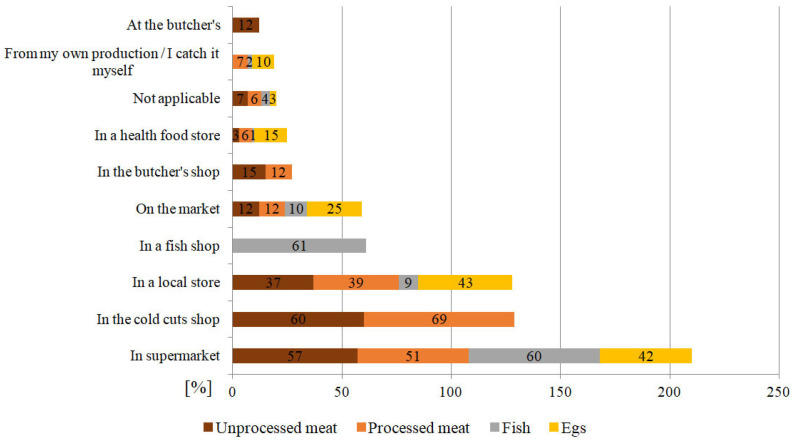

Figure 2Respondents’ preferences of eggs selection [%] regarding hen rising method.
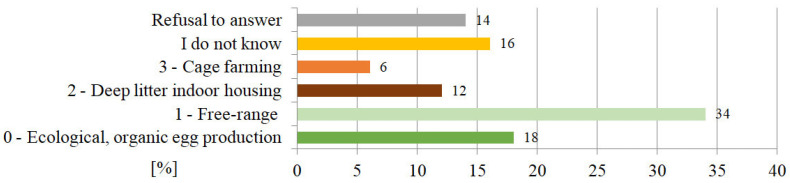



### 3.3. The Characteristic of the Trends in the Consumption of Animal Products

Our respondents, who in majority were highly educated people, when supplying their households with eggs, chose only hen eggs. 16% of the participants generally did not pay attention to the egg marking that allowed consumers to distinguish free range eggs and organic farming eggs from the industrial caged hen production (Figure 2). The largest number of respondents bought eggs marked as 1) free range eggs, (34%), then 0 organic egg production (18%), 2) deep litter indoor housing (12%), and finally 3) cage farming (6%). There were no significant differences in the responses of the respondents representing different socioeconomic groups.

#### 3.3.1. The Frequency of the Consumption of Animal Products

Among the examined animal products, processed meat (e.g., cold-cuts, sausages, and pâtés) was consumed daily by 5% of the respondents, unprocessed pork by 2%, and eggs by 4% of the participants. In terms of unprocessed meat, the following types were consumed most frequently by Polish respondents (in descending order): chicken > pork > turkey > duck > beef > veal > lamb. Consumers, who declared the consumption of animal products several times a week (from 1 to 6 times a week) most often purchased: eggs (79%), processed meat (58%), pork (unprocessed meat) (49%), beef and chicken (24% each) and fish (38%). Among people declaring consumption of animal products “several times a month”, 37% of respondents chose fish, 25% turkey, and 19% beef. The consumption of animal products “several times a month” was declared by 9% of respondents who consumed processed meat. Unprocessed meat which was eaten the least frequently (answers “a dozen times a year” and “several times a year”, respectively) included: goose (30% and 6%), duck (35% and 15%), mutton (15% and 6%) and lamb (15% and 4%). Consuming eggs twice a week was declared by 30% of the respondents, 3 times a week by 21%, several times a month by 13%, once a week by 12%, and 4 times a week by 7%. Detailed answers are presented in Figure 3.

#### 3.3.2. The Quantity of the Consumption of Animal Products at One-Serving

In terms of the amount of animal products consumed at one time, the study revealed that unprocessed meat was most often consumed in 200 g portions (duck numbered 51% of indications, veal, and pork 47% each, goose 46%, beef 41%, chicken and turkey 40% each, mutton 33%, and lamb 29%). They were followed by those who ate 100 g of meat portions (beef, veal, mutton, and lamb 33% were of the answers in each case, pork and chicken 32% each, turkey 28%, duck 19%, and goose 17%). The 300 g portions were listed third for the following unprocessed meat: beef, pork, lamb, chicken, and goose. Respondents who ate 50 g portions ate the following unprocessed meats: veal, mutton, lamb, turkey, duck, and fish. In the case of processed meats (cold cuts, sausages, pâtés), the respondents most often chose the following portion sizes for a one-time consumption: 2 slices (30% of indications), 3 slices (25%), 4 slices (18%), 50 g (17%). In terms of the consumption of hen eggs, respondents most often ate the following amounts during one-time consumption: 2 pieces (55% of respondents), 1 piece (33%), and 3 pieces (6%). Detailed results are presented in Figure 4.

#### 3.3.3. Statistical Analysis Related with Egg Marking

The Chi-squared test was performed to analyze the obtained results related with egg marking. The *p*-values obtained for the analyzed socio-demographic features, in particular, sex (Table 2), marital status (Table 3), net income level (Table 4), and education level (Table 5) were all higher than 0.05 confirming the null hypothesis describing that none of these features had impact on choosing the egg marking among investigated respondents.

## 4. Discussion

Our surveys were conducted in 2017, however the discussion section was performed based on the relevant literature mostly after 2020. When the data were missing, we looked for the older data related. Unfortunately, the related research on the topic of sustainability among Polish people were not so common among scientists [13,22,39,40,41]. For instance, research on the consumption of meat in Poland, published in 2022, concerning, inter alia, wild game meat, was not mentioned by our respondents, therefore it was not included in the discussion. Moreover, wild game meat consumption [11], that in not sustainable nor popular among wider group of consumers in Poland.

As already underlined, the number of responds received was much lower than expected. Due to the very limited number of answers as well as that the respondents were mainly classified as highly educated people, the statistical analysis like factor analysis and cluster analysis did not give statistically significant results. Therefore, our results constitute the preliminary research on sustainable consumption conscious performed primarily among people, who based on their socioeconomic status should reveal the highest awareness on balanced and sustainable diet, as education level is correlated with the income level.

Moreover, due to the number of responses received, our research appeared qualitative rather than quantitative, thus they could not be treated as describing the trend in the whole Polish population. Thus, this fact was strongly underlined and during showing the results we put the emphasis that the received percentages apply mainly to this group of respondents.
Figure 3The frequency of animal products’ consumption [%] among respondents.
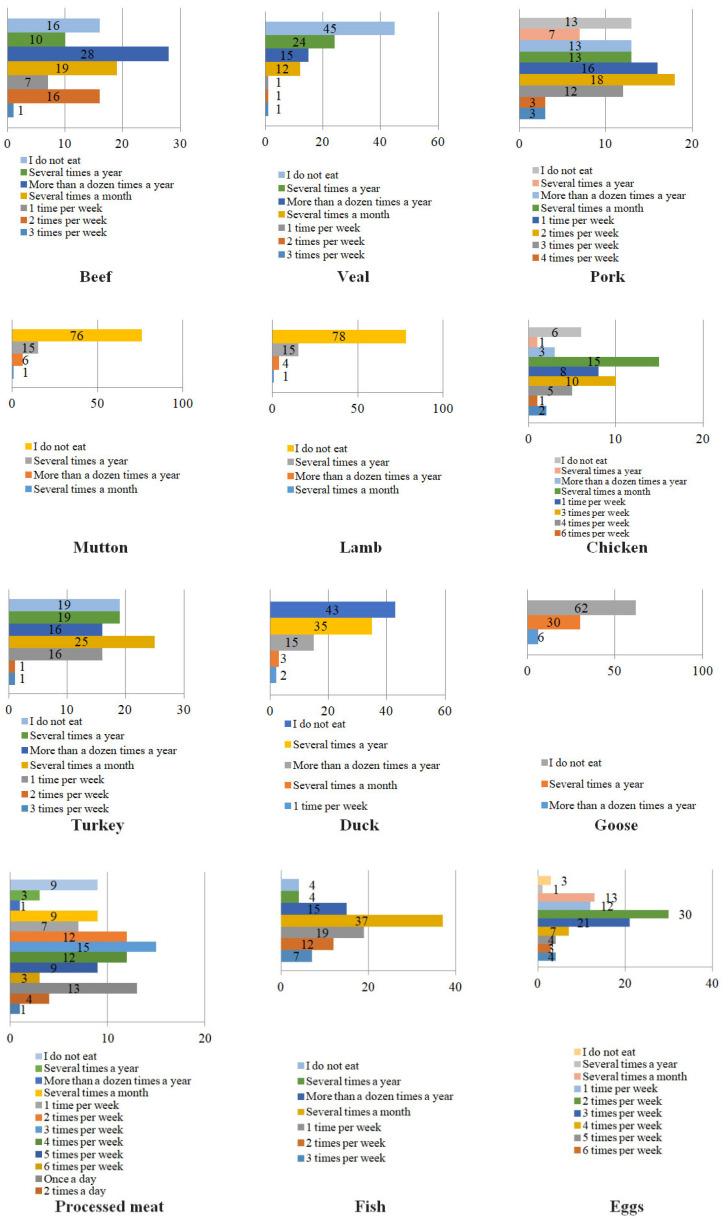

Figure 4The amount of animal product consumed at once [%].
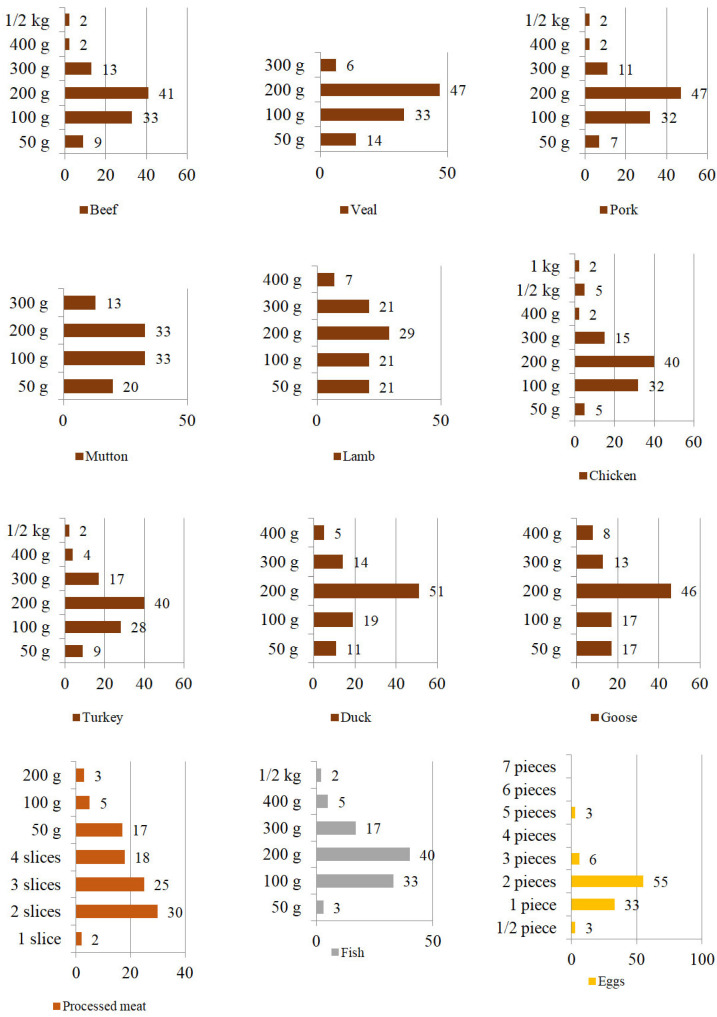

ijerph-19-13072-t002_Table 2Table 2The egg marking choices according to the sex of the respondents.Egg MarkingObservedExpected*p*-ValueWomenMenWomenMen31029.31032.68970.2674218517.84485.15521716.20691.79310403.10340.8966?658.53452.4655?—I do not know.
ijerph-19-13072-t003_Table 3Table 3The egg marking choices according to the marital status of the respondents.Egg MarkingObservedExpected*p*-ValueSingleMarried/in RelationSeparation/After DivorceWidowedSingleMarried/in RelationSeparation/After DivorceWidowed329013.64297.07140.42860.8570.08032616006.678612.96430.78571.5714132022.1254.1250.250.5012101.21432.35710.14290.2857?54113.33936.48210.39290.7857?—I do not know.
ijerph-19-13072-t004_Table 4Table 4The egg marking choices according to the indicative net income **(PLN)** of the respondents.Egg MarkingObservedExpected*p*-Value<1000 PLN 1001–3000 PLN3001–5000 PLN5001–7000 PLN>9001 PLN<1000 PLN 1001–3000 PLN3001–5000 PLN5001–7000 PLN>9001 PLN3022320.39134.69571.95651.36960.58700.34682086310.78269.39133.91302.73911.17391141000.26093.13041.30430.91300.39130130000.17392.08700.86960.60870.2609?071100.39134.69571.95651.36960.5870?—I do not know.
ijerph-19-13072-t005_Table 5Table 5The egg marking choices according to the educational level of the respondents.Egg MarkingObservedExpected*p*-ValueSecondary EducationSecondary VocationalPost-SecondaryHigher VocationalBachelor’s DegreeMaster’s DegreeSecondary EducationSecondary VocationalPost-SecondaryHigher VocationalBachelor’s DegreeMaster’s Degree300200100.87270.43641.09090.43640.87278.29090.7677210222141.52730.76361.90910.76361.527314.509111100140.50910.25450.63640.25450.50914.836401000030.29090.14550.36360.14550.29092.7636?1110170.80.410.40.87.6?—I do not know.


Based on these qualitative results received, it was observed that the awareness of the sustainable consumption was not at the high level among the group of people who, based on their economic and knowledge predisposition, are expected to have the highest understanding and application rate of the sustainability rules in the real life. Based on our preliminary results, we could conclude that other groups of people with lower educational level and related income level were on the much lower level of awareness. The results also implied that further studies on larger scale would be justified and necessary. The European projects on consumption rates (i.e., EU Menu project) or related national projects in particular member countries of the EU do not include additional questions on the perception and conditions of certain consumption behaviors. Moreover, as we already assumed that the general level of sustainability of consumption among Polish respondent will be low, further research could also contain the section of questions regarding factors that respondents perceive as necessary and required during the transformation process from unsustainable to sustainable consumption regarding the European Union sustainable development goals. As the Earth ecosystem cannot lift current consumption style for much longer, effective changes to sustainable consumption are necessary regardless of if it is required by the society or not. Thus, our preliminary and further results if continued will bring new insights in the practical aspects of transformation process.

The research results, however, possessed on the small group of respondents, might serve for the wider audience as follows. Individuals (private persons) might be interested in trends in healthy eating and sustainable consumption. Moreover, those people who are forced by their own health conditions or by their relatives to seek information about the possibilities and guidelines for more balanced nutrition. Food business entities, both physical and legal, might use survey results for paying attention to safety aspects and to protect the interests of consumers in order to ensure human health and improve the quality of life. Finally, entities operating in the education sector might use the results in order to better inform and develop healthy consumption habits in society, conducting research and training to promote sustainable practices and carrying out activities aimed at avoiding or reducing the risk related to the consumption of animal products.

### 4.1. Meat Consumption

The researchers discuss sustainable meat consumption either in terms of its impact on the individual psychophysical condition or the natural environment. Undoubtedly, meat is still one of the most important foods in the human diet. However, in comparison to other animal products, meat is also the most controversial product [42]. This is not only due to the fact that the demand for meat has been growing [43,44], but also because social awareness of the environmental costs of producing and preparing meat meals has been increasing. Household income has the greatest impact on the amount of red meat and preserves consumed [45]. However, the research of Whitton et. al. [46] showed that after reaching a certain level of prosperity (that is US$40 thousand GDP per capita), meat consumption no longer increases.

Sustainable consumption of meat includes maintaining a rational and healthy level of consumption while minimizing the damaging effect of meat production on ecosystems and the welfare of farm animals [47]. The role of consumers oriented towards sustainable meat consumption is driven by their choices which are the result of sustainable production and changes in one’s own habits. Taking into account the aspect of sustainability, meat consumers face the following choice: consumption at the same level as before, paying particular attention to the selection of products, limiting the consumption of meat, or no longer eating it altogether.

Eating meat can have both a positive and negative effect on human health. Meat is a rich source of energy and nutrients, such as proteins, microelements, vitamins, and bioactive substances, like L-carnitine, creatine, carnosine, anserine, taurine, conjugated linoleic acid, α-lipoic acid, coenzyme Q10, γ-aminobutyric acid, glutathione, and bioactive peptides [48]. However, it also contains saturated fat and cholesterol [49]. The results of research by Montoro-García et al. [50] showed that the regular consumption of pork dry-cured ham improves systolic/diastolic blood pressure and facilitates the maintenance of metabolic pathways, which may be beneficial in the prevention of cardiovascular diseases. Research by Abrhaley and Leta [51] has shown that camel meat is a good source of nutrients, both in terms of composition (including low fat and cholesterol content) and declared health effects, which is especially evident in Somalia and India. In addition, the complete elimination of meat from the diet may worsen mental health, in particular causing depression and anxiety [52]. A future solution of cultured meat [53], also known as slaughter-free, lab-grown, or synthetic meat seems inevitable. This is meat produced by in vitro cell cultures of animal cells [54]. The type of meat and its source e.g., due to the adulteration of meat products [55], as well as the form of its processing are also important. The most controversial health aspect of meat consumption is related to the likely carcinogenic effects of red and processed meat [56,57], as well as the risk of ischemic heart disease [58].

In addition, the amount and type of additives used during meat processing also raise concerns. However, additives are used due to customers’ expectations regarding a food that is durable, safe, and sensually attractive, the optimization of production costs and other factors, such as variability of the quality of the raw material, and changes in legal regulations, etc. Some additives (for example, sodium nitrate) are used more often than others [59]. Not all additives may affect all consumers, and in some cases, children may be particularly vulnerable [60]. Concerns about meat consumption also refer to such threats as: the legacy of the avian influenza epidemic, mad cow disease, genetic modification, bacterial infections, and the use of antibiotics and pesticides [61]. Research conducted in Switzerland [62] showed that consumers’ concerns about the presence of hormones and antibiotics in meat are greater than the potential negative effects of fat and cholesterol. Although the results of the research by Salejda et al. [63] indicated that young Poles read food labels, paying particular attention to the content of preservatives, they do not always correctly interpret them.

The same research indicated that most often Polish respondents bought meat at a butcher’s and in meat shops: unprocessed meat (60% of the answers), processed meat (69% of the responses). Secondly, the respondents bought meat at meat stands in supermarkets: 57% and 51% of responses, respectively. These places earned consumers’ trust and were quite popular: butcher shops (15% and 12%), markets (15% and 12%), as well as slaughterhouses (12%—unprocessed meats). Polish respondents did not associate health food stores with meat products. Only 3% of those who consume unprocessed meat and 6% of those who eat processed meat bought it in health food stores. For some respondents, the convenience and the proximity of the store might be more important than the quality of meat, particularly for those who bought meat in local stores, respectively 37% and 39% of responses. These results might also be affected by increasing governmental actions, i.e., legal requirements (food law regulations on production processes in terms of food safety and quality) and the quality of meat, regardless of the type of distribution facility. The quality is one of the key factors in buying meat. According to consumers, the quality of meat was closely related to its appearance (including the color and noticeable fat), as well as nutritional and dietary values [40]. The quality is also affected by the freshness of meat, and not only the amount of fat but also its distribution [64].

In terms of choosing specific types of meat, limiting the consumption of certain types of meat or diversifying the diet by replacing it with other types of meat, our research showed that red meat (pork, beef, veal, and lamb) was consumed more often than white meat (i.e., poultry and rabbit). As for red meat, pork was consumed most often: 49% of the respondents declared its consumption 1 to 4 times a week, and 3% of the respondents eat it every day. Lamb (several or a dozen times a year: 21% of indications) and mutton (a few or several times a year: 19%) were the least consumed red meat. In terms of white meat, hen meat was consumed most: 24% of the respondents declared eating it 1–6 times a week, and 2% of the respondents consumed it every day. Goose meat was the least popular (several or several times a year—36% of responses). The consumption of processed meat was very popular. More than half of the respondents (55%) consumed it from 1 to 5 times a week, and almost every fifth person (18%) from 1 to 3 times a day.

The evident popularity of chicken meat is in line with global tendencies, which show that poultry replaces other meats [46]. In Poland, especially in the period from 1994 to 2014, there was an increase in the production of poultry meat [65]. According to the same source, the increase was caused by its popularity, which was triggered by its relatively lower price and health benefits. Supply factors include a short production cycle, lower production costs than in other EU countries, and orders placed by other EU member states after Poland joined the EU [64]. However, lowering the price of white meat (especially poultry) in comparison to red meat does not always increase its consumption [45]. Poultry and sheep farming play a significant role in improving the environment, resources, and biodiversity. Apart from the taste and nutritional value, the consumption of mutton meat can also be encouraged by curiosity. Organizing tasting sessions and gastronomic events may be helpful in promoting the consumption of sheep meat [66].

Limiting meat consumption by replacing it with fish or a plant-based diet is associated with the concept known as flexitarianism, which is defined as an occasional refraining from eating meat. The study by Sijtsema et al. [9] showed that this is significantly dependent on a consumer’s personal motivation, nutritional knowledge, the ability to prepare meatless meals, social and physical support, i.e., the availability of meat substitutes, their taste, the convenience of eating substitutes, or the reaction and support of the immediate environment, (e.g., relatives). The research by Zur and Klöckner [67] narrowed the motivation for meat consumption to three groups of factors: moral (animal and human rights), health and environmental (overproduction of greenhouse gases, and anthropogenic ammonia (depletion and pollution of water resources and loss of biodiversity). On the other hand, meat consumption has been justified by harmlessness (the more meat is consumed, the perception of environmental disadvantages decreases [68,69], purposefulness (the belief in the rationality of one’s preferences [68,70], culture (typical of a given social group), and economy (meat sales revenues [68,70]. According to Western culture, a meat diet is viewed as typically male, while vegan and vegetarian diets as more female [71,72]. Eating meat may also be a way of expressing social identity, e.g., prestige or style [73,74] which has also been present throughout history, namely the rationality of consumption, fear of fat, vegetarian philosophy, or loss of trust. According to a relatively new social tendency, one should refrain from consuming meat due to the negative impact of its production on the environment. The research by Kucharska and Borusiak [41] showed that especially young consumers perceived the negative impact of industrial meat production on the environment and believed that limiting meat consumption could improve the condition of the natural environment. In addition, 40% of respondents participating in this study intended to limit meat consumption precisely for environmental reasons. However, the difficulties in limiting meat consumption may include the inability to compose a balanced diet and the lack of time to prepare balanced meals [41]. However, promoting the reduction of meat consumption among adolescents requires overcoming the belief that eating meat is not only pleasant but also, normal, natural, and necessary [75]. According to a French study [76], the decline in meat consumption may be caused by concerns about its impact on one’s health and the natural environment (including animal welfare). On the other hand, research conducted in Iraqi Kurdistan [77] showed the opposite tendency, indicating that animal welfare does not have a statistically significant effect on the consumption of any type of meat. In turn, studies conducted in Spain [78] showed that the risk of cancer and increased mortality resulting from meat consumption also does not significantly reduce its consumption. This is especially true of men, who are much less likely to reduce their meat consumption than women. Our research showed that meat was much more popular in the diets of Poles than fish. According to the data provided by the Polish Central Statistical Office [27] (pp. 140–141, 182–183, 319) in 2019, the average monthly consumption of meat per capita in households was 5.08 kg/person, while fish and seafood 0.27 kg/person. Moreover, according to the same data [27] (pp. 140–141, 182–183, 319), in 2019 on average, 5.08 kg of meat consumed per person in the Polish household included 2.87 kg of raw meat, 1.53 kg of poultry, and 1.97 kg of cold cuts and processed meat. In terms of socioeconomic groups, the most meat was purchased by retirees and pensioners (6.77 kg), as well as farmers (5.83 kg).

Another way of implementing sustainable activities into daily life is giving up consumption—abstinence from eating meat and switching to vegetarianism or veganism. Another example of animal products that are a rich source of nutrients (especially proteins) are the edible insects consumed by the inhabitants of Asia, Africa, and South America. Insect based foods are a sustainable meat substitute [79,80,81]. They may affect both food safety and sustainable development. However, much of this depends on nutritionists’ approach [10] and the development of insect farming technology [82,83]. The informational aspect is also crucially important, especially in cultures where insects are considered to be pests. Due to the above, it is also important to verify consumers’ knowledge. For example, the study by Guine et al. [84] shows that although the knowledge about the sustainability and consumption of insects in some countries is quite common, many consumers still rely on misconceptions. Therefore, this is significantly dependent on the national educational strategies which would be based on such direct motivational techniques as encouraging tasting. In our study, participants did not include insects into edible animal products. Furthermore, all respondents declared meat consumption, so there were no vegetarians or vegans among them. Moreover, meat products are more popular than fish, which also relates to RQ1. According to the respondents, health food stores were not associated with the place where meat was obtained, and the proximity of stores to the place of residence was often more important than the aspect of its quality, which is a reference to RQ2. The results of our study showed a slight shift towards sustainability in meat consumption among Polish respondents, who in our studies were mainly people with higher education. Therefore, educating Poles about both the harmful and beneficial properties of meat and the variety of non-meat substitutes is necessary.

### 4.2. The Consumption of Fish Products

The research on balanced fish consumption has been discussed mainly in terms of a balanced diet and recommendations promoting fish consumption in a healthy and balanced diet. This includes enumerating the numerous advantages of fish such as variety of nutrients (a source of protein, vitamins, minerals, and unsaturated fatty acids) and their positive effects on the psychophysical human condition, that prevent and alleviate both civilizing and age-related diseases [14]. Eating fish can also have a positive effect on the brain, immune, and cardiovascular systems [14], as well as counteract dementia and intellectual disability [15]. Regular consumption of fish may significantly improve a person’s quality of life and decrease multiple sclerosis and other disabilities [16]. Consuming fish (especially sardines) may prevent a person from developing metabolic diseases, such as type 2 diabetes [85]. Daily consumption of a nutritionally balanced meal which includes soy and fish reduces the risk of developing lifestyle diseases [86]. However, the positive effect of fish on the human body depends on the species of fish and the water (area) the fish originate from. They may contain too many harmful substances, e.g., arsenic [17], mercury [87] or selenium [88], and thus are harmful to human health. This is especially important for women who are or who are planning to be pregnant [89]. Therefore, fish offered for sale should transparently show where they were caught. According to the Polish Central Statistical Office ([27] (pp. 140–141, 182–183, 319), the average consumption of fish and seafood per person in a typical Polish household in 2019 was 0.27 kg. On average, Poles spent PLN9.90 [US$2.2] on fish and seafood monthly. In terms of sociodemographic groups, fish and seafood were most popular among the oldest members of the society, i.e., among retirees and pensioners, who on average consumed 0.40 kg of fish per month [27] (pp. 140–141, 182–183, 319).

In the researched group of Polish respondents, fish were not selected very often. The largest group of participants consumed fish only a few times a month (37%), once a week (19%) or several times a year (12%). Only 7% of the respondents ate fish three times a week. This indicates the need for educational activities which would encourage consumption of fish products in order to better balance the daily diet of Poles. This is consistent with other research findings [90], which underline the unsatisfactory consumption of fish in Poland. The research also provided additional information on the fish preferences of Poles, who mainly prefered pollock, cod, carp, trout, river cobbler, and salmon. Among processed smoked fish products, mackerel and salmon were the most common, and tuna, sprats, sardines, and herring were the most popular canned products. However, the authors emphasize the considerable potential of increasing the demand for fish in Poland. It depends on how the fish industry will modernize and whether it will become more physically and economically available. Educational activities promoting proper fish handling techniques to reduce the waste may also increase fish consumption [91].

The factors affecting the inclusion or elimination of fish in the diet, other than substances harmful to health, also include: difficulties in acquiring and eating (convenience aspect), cost-effectiveness [92] including the price [7], no habit of eating fish [7], and a willingness to change the menu [90]. Their specific smell and the presence of bones may also prevent young people from eating fish. Therefore, when preparing a meal, it is recommended to pay attention not only to its taste, but also to the smell and appearance. This also applies to the selection of sauce and potatoes (or their equivalents, such as rice, porridge, or pasta), as well as educating young people on how to prepare meals [93]. Research conducted in Hungary by Temesi et al. [94] indicated that a preference for fish is not affected by the fish preferences of other household members. However, it may be affected by the ability to prepare them. Therefore, food policies should focus primarily on improving cooking skills [95].

Fish consumers in Poland can also be grouped into those who are and are not aware of health benefits of eating fish. The first group is mainly represented by elderly and highly educated people [96]. The underestimation of fish by Poles is also emphasized in the research of Kosicka-Gębska and Ładecka [97], who discussed general consumption patterns and the fact that in Poland, fish is most popular during fasts and holidays. Other researchers also pointed out the need for campaigns promoting the health benefits of eating fish [98]. They also emphasized the importance of reliable and objective information on benefits of eating fish, as well as a list of specific species that are essential in a balanced diet [99]. 

The conducted research showed that most often fish was bought in fish shops (61%) and supermarkets (60%). Only 1% of respondents purchased fish in health food stores. This may result from the fact that fish shops offer a better selection of fish, not only in terms of variety, but also the form (e.g., fresh, chilled, frozen, dried, smoked, etc.). However, fish shops are not always preferred nationwide. For example, despite the fact that they preferred fresh fish, most often, Bulgarian consumers bought fish in supermarkets [7]. The respondents quite rarely chose fish for their meals and did not obtain it in health food stores, which is a reference to RQ1 and RQ2.

### 4.3. The Consumption of Eggs

In terms of balanced egg consumption, similarly to the other discussed food products, researchers have discussed it mainly in terms of a balanced diet and eating recommendations. The main benefits of eating eggs include nutritional value, low caloric content, low price, easy preparation, versatile use, and convenience [18]. In terms of the latter researchers, there are some doubts due to the negative effects of high cholesterol intake. It is worth emphasizing that hen eggs are the most popular type of eggs, in Poland. Some researchers discussed the difficulties in buying eggs of other birds, including turkeys and ducklings [19]. Although quail eggs are smaller than hen eggs, they contain more cholesterol, so it is recommended to consume them with caution [20]. In terms of hen eggs, some research suggested that avoiding their consumption may pose a greater threat to the human condition than eating them, while other studies recommend eating them with caution. Much depends on the right proportions of the consumed products, as well as the overall health condition of consumers. Some of the egg nutrients are essential during pregnancy, infancy, and early childhood due to the content of iodine, choline, and DHA unsaturated acids [100]. Higher egg consumption may also lower the risk of multiple sclerosis [21]. Research conducted in China by Xia et al. [101] indicated that both low and high egg consumption may lead to cardiovascular disease, therefore, it is suggested to consume less than 6 eggs per week. Other research on the metabolic syndrome—a major risk factor for cardiovascular disease—carried out in Korea by Park et al. [102], found that eating 4–7 eggs a week may lower the risk of metabolic syndrome, and eating two or more eggs a day does not reduce this risk in adult Koreans. Another study conducted in China by Ji et al. [103] found no significant differences in arterial stiffness caused by egg consumption. However, this research also showed that moderate egg consumption (3–3.9 eggs/week) may have a beneficial effect on stiffness.

According to the Statistics Poland [27] (pp. 140–141, 182–183, 319), the average consumption of eggs per person in a Polish household in 2019 was 10.99 eggs. The average monthly cost of eggs per person was PLN6.89 [US$1.53]. In terms of socioeconomic analysis, the largest number of eggs in 2019 was consumed by retirees and pensioners (on average 14.81 eggs/person) and farmers (11.80 eggs/person). During the COVID-19 pandemic, due to the closure of hotels and catering establishments in Poland, the consumption of eggs increased, and a further increase in consumption of dairy and eggs has been expected [104]. Our research results showed that most of the respondents consumed between 1 and 3 eggs a week, 14% of them consumed eggs to a small extent (several times a year and several times a month), and 53% ate them in moderation (2–5 eggs a week). These results indicated a need to educate the Polish society about the recommended consumption of eggs and their nutritional value. The need to promote the nutritional benefits of eating eggs, as well as correct misconceptions about the negative impact of cholesterol in eggs, which is also emphasized in research by Talakesh et al. [105]. 

The majority of Poles surveyed in our research (70%) paid attention to the marking of eggs and the type of poultry farming. Most of these respondents declared that they buy eggs from free range (34%) and organic farming (18%). Therefore, this aspect of egg consumption can be described as sustainable. However, knowledge about the sustainability in terms of egg consumption varies nationwide. For example, the research by Mizrak et al. [19] carried out in Turkey indicated that 72% of families did not have this knowledge. On the other hand, Italian respondents paid much more attention to the marking of eggs [19]. One solution for maintaining sustainability is to keep food consumption constant by choosing sustainable substitutes (e.g., eggs of other birds or plant products). The research by Kralik et al. [106] also showed that female consumers paid more attention to the shelf life of eggs, their storage, cholesterol and fat content, shell damage, and egg soil than male consumers [106].

Sustainability may also be discussed in terms of buying eggs in health food stores. However, the results of our research did not show such a tendency. Only 15% of respondents bought eggs in organic food stores. Another aspect of sustainable egg consumption is transport. Our research showed that eggs were usually purchased in a local store (43%), in a supermarket (42%) or at a market (25%). Similar results were revealed in the research conducted by Prencipe et al. [107] in Italy. Italian respondents also bought eggs from large retailers (53%), small retailers (25%), direct producers (16%), and local markets (6%). Also, research by Mizrak et al. [19] showed that the majority of respondents (68%) bought eggs in supermarkets. This shows that sustainable solutions are not used when purchasing local products. With regard to transport, no activities minimizing the need for long distance transport were observed.

The slight consumption of eggs and the fact that the eggs were not obtained by our respondents in health food stores also indicated the refutation of RQ1 and RQ2. However, the optimistic trend noticed was, that almost 3/4 of our respondents paid attention to the marking of eggs from the point of view of poultry farming, choosing free range eggs most often.

Based on the results of Chi-squared test performed for question related with egg marking and sociodemographic studies it was revealed that neither sex, marital status, income level, nor education level had the statistically significant impact (*p* < 0.05) on the choice of respondents in our questionnaire studies on choosing the marking of the bought eggs. 

### 4.4. Limitations of the Study

The main limitation of our study was related with the rate and the sociodemographic characteristic of the received questionnaires. The difficulty in estimating the number of respondents was, inter alia, no information on the number of adult Poles with higher education who were consuming meat, fish, and eggs were in fact responsible in the household for purchasing animal products; they were not sharing household obligations related to the purchase of animal products.

We have collected only 67 full questionnaires, that made any statistical assessment not significant in terms of for the Polish society. In our opinion, the method of spreading the questionnaire was mostly relying on the free will of the respondents to invest their free time to provide the answers to in their opinion boring questions without any gratification. It leads us to the presumption that while performing such questionnaires is the form of some gratification for the respondent, it might have the impact on the number of answers received, as well as for the differentiation in the sociodemographic status of the respondents. 

Related limitation to the above was that majority of the respondents in our study were adults having the high education level. It affected the general answers, and our conclusions can be made only for this group of inhabitants. On the other hand, the results showed that if the consumption pattern of animal products were unsustainable among the highly educated participants, having the knowledge and economical possibility to choose a healthy, sustainable diet still is the most expensive form of nutrition in Poland. Thus, much lower level of sustainability would be expected to be received if there were answers from the participants from other socioeconomic groups. 

The next issue was the fact that the research was based on the voluntary declarations rather than the actual observation of the respondents’ behaviors. It is uncertain if the provided information was the same as the decisions made by the respondents. Therefore, this research should be continued, and could be complemented with food diaries [8]. The follow-up study should efficiently address a larger group of respondents and include the equal representation of respondents representing all sociodemographic groups according to sex, age, level of education, residence, number of people in a household, and income range. It could also include a wider group of animal products, such as goat, horse, and game meat, farmed pheasants, guinea fowl, and others, a wider range of dairy products, the eggs of birds other than hens, insects, crustaceans, and seafood intended for human consumption. In terms of processed meat, the study could explore specific types of it and the form of consumption (raw, partially cooked, cooked), packing methods, and storage methods used between purchase and final consumption. Our study included not only closed- but also open-ended questions about other consumed animal products and their quantity. However, the respondents were not proactive and responded mainly to closed-ended questions. Therefore, a future follow-up study should be designed based on a closed list of choices. A variety of closed questions seems to be a better tool for obtaining reliable results.

The discussion on the limitations presented above points that continuing the research in the future on larger scale and with improved questionnaire survey with performing statistical analysis on the results received would allow to analyze changes and reasons of the eating preferences of the respondents in the general population. Although the discussed research results were limited, they still may provide a valuable insight into the opinions and attitudes of Polish consumers regarding the consumption of selected animal products. The results of this type of research are essential in the risk assessment analysis related with consumption exposure.

## 5. Conclusions

The results of the conducted research provided preliminary significant insight into the sustainable consumption of animal products in Poland. It was stated that the sustainability of the animal product consumption among the group of people with the highest education level, correlated with the high-income rate, was low. Thus, among other groups of citizens with lower educational level and income, it is expected to be even lower. Regarding the necessity of transformation towards sustainability in the European Union and its goals, our research revealed the need to extend the research to hear the voice of different sociodemographic groups to make this transformation as fast and as effective as possible. They also contributed to the promotion of sustainable development and a greater understanding of consumer behaviors with regard to (both processed and unprocessed) meat, fish, and eggs. The results of the surveys have revealed that among our group of respondents, all declared consuming animal products. No significant patterns between socioeconomic aspects and respondent choices were observed. Our surveys reveled that egg consumption was the most sustainable among the investigated animal products. However, for all the analyzed animal products, unsustainable consumption patterns were stated. Among respondents who consumed animal products several times a week, the most frequently chosen in decreasing order were eggs, processed meat, pork, beef and chicken, and fish products. Fish products were consumed by most of the respondents several times a month and once a week. Most respondents declared the consumption of hen eggs several times a week. The majority of them also paid attention to egg coding describing the hen rising method, choosing mainly free-range eggs (code 1). The results of our survey may be followed up by more detailed research in this regard. Furthermore, the study clearly indicated the need for education, motivation, and encouragement in decreasing consumption of animal products and better balancing the daily diet of Poles.

## Data Availability

Data available on request due to restrictions eg privacy or ethical.

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
