# Peer review of "Sustainable or Not? Insights on the Consumption of Animal Products in Poland"

_ijerph, 2022, doi:10.3390/ijerph192013072_

Round 1
Reviewer 1 Report
It is an article of hight current interest. The population in Poland is 40 million, they have a very low n, only 67 answers that hardly allow an adequate inference, with external validity. On the other hand, the research is very much aimed at people with a high academic level (67% master's degree), which is a bias. The summary should clarify the limitation of the sample size. The above can be adjusted, avoiding generalizing, focusing its results on the segment. I really like how the results are presented.
Author Response
Review 1
(x) I would not like to sign my review report
( ) I would like to sign my review report
English language and style
( ) Extensive editing of English language and style required
( ) Moderate English changes required
(x) English language and style are fine/minor spell check required
( ) I don't feel qualified to judge about the English language and style
|
Yes |
Can be improved |
Must be improved |
Not applicable |
|
|
Does the introduction provide sufficient background and include all relevant references? |
(x) |
( ) |
( ) |
( ) |
|
Are all the cited references relevant to the research? |
(x) |
( ) |
( ) |
( ) |
|
Is the research design appropriate? |
( ) |
( ) |
(x) |
( ) |
|
Are the methods adequately described? |
( ) |
(x) |
( ) |
( ) |
|
Are the results clearly presented? |
( ) |
(x) |
( ) |
( ) |
|
Are the conclusions supported by the results? |
( ) |
( ) |
(x) |
( ) |
Comments and Suggestions for Authors
It is an article of hight current interest. The population in Poland is 40 million, they have a very low n, only 67 answers that hardly allow an adequate inference, with external validity. On the other hand, the research is very much aimed at people with a high academic level (67% master's degree), which is a bias. The summary should clarify the limitation of the sample size. The above can be adjusted, avoiding generalizing, focusing its results on the segment. I really like how the results are presented.
Submission Date
23 August 2022
Date of this review
02 Sep 2022 15:52:15
We would like to thank the Reviewer for the constructive and valuable comments. The detailed responses are placed in the relevant places below.
It is an article of hight current interest.
The population in Poland is 40 million, they have a very low n, only 67 answers that hardly allow an adequate inference, with external validity. On the other hand, the research is very much aimed at people with a high academic level (67% master's degree), which is a bias. The summary should clarify the limitation of the sample size. The above can be adjusted, avoiding generalizing, focusing its results on the segment.
Thank you for the valuable comment. Based on the Reviewer’s comment we have added the following paragraphs in the results, discussion, and conclusion sections. Please see the revised version of the manuscript in lines: 197-198, 213-217, 219-220, 242-243, 283-290, 496-498, 628-644, 663-666.
I really like how the results are presented.
Reviewer 2 Report
The paper deals with meat consumption.
Hypotheses and hypotheses development are missing. The literature review is lacking. The study applies more than 100 sources. The results are compared with previous sources, but these must be explained also in the literature review, as the basis of the research.
Figure 1. is unnecessary. It is enough to refer to those points of SDG that are relevant in the study.
The applied methods are not explained according to the secondary literature. Factor analysis and cluster analysis are recommended.
I would suggest applying a theoretical framework. There are a lot of models that are connected with meat consumption.
The figures about the results reflect only which meat type and how often it was consumed. These data could have been gained from the national statistical office or Eurostat. Other questions are more important, such as which factors determine meat consumption.
The sample size is very low. Thus, this study is rather qualitative. At this low sample size, it is not allowed to apply percentages.
Author Response
Review 2
(x) I would not like to sign my review report
( ) I would like to sign my review report
English language and style
( ) Extensive editing of English language and style required
(x) Moderate English changes required
( ) English language and style are fine/minor spell check required
( ) I don't feel qualified to judge about the English language and style
|
Yes |
Can be improved |
Must be improved |
Not applicable |
|
|
Does the introduction provide sufficient background and include all relevant references? |
( ) |
( ) |
(x) |
( ) |
|
Are all the cited references relevant to the research? |
(x) |
( ) |
( ) |
( ) |
|
Is the research design appropriate? |
( ) |
( ) |
(x) |
( ) |
|
Are the methods adequately described? |
( ) |
( ) |
(x) |
( ) |
|
Are the results clearly presented? |
( ) |
( ) |
(x) |
( ) |
|
Are the conclusions supported by the results? |
( ) |
( ) |
(x) |
( ) |
Comments and Suggestions for Authors
The paper deals with meat consumption.
Hypotheses and hypotheses development are missing. The literature review is lacking. The study applies more than 100 sources. The results are compared with previous sources, but these must be explained also in the literature review, as the basis of the research.
Figure 1. is unnecessary. It is enough to refer to those points of SDG that are relevant in the study.
The applied methods are not explained according to the secondary literature. Factor analysis and cluster analysis are recommended.
I would suggest applying a theoretical framework. There are a lot of models that are connected with meat consumption.
The figures about the results reflect only which meat type and how often it was consumed. These data could have been gained from the national statistical office or Eurostat. Other questions are more important, such as which factors determine meat consumption.
The sample size is very low. Thus, this study is rather qualitative. At this low sample size, it is not allowed to apply percentages.
Submission Date
23 August 2022
Date of this review
05 Sep 2022 19:53:25
We would like to thank the Reviewer for the constructive and valuable comments. The detailed responses are placed in the relevant places below.
The paper deals with meat consumption.
Hypotheses and hypotheses development are missing.
Thank you for the valuable comment. The added paragraph is contained in lines 119-124, 672-680.
The literature review is lacking. The study applies more than 100 sources. The results are compared with previous sources, but these must be explained also in the literature review, as the basis of the research.
Thank you for the comment. The added paragraph is contained in lines 56-66.
Figure 1. is unnecessary. It is enough to refer to those points of SDG that are relevant in the study.
Thank you for the valuable comment. Figure 1 has been removed.
The applied methods are not explained according to the secondary literature. Factor analysis and cluster analysis are recommended.
Thank you for your very important comment. Due to the number of data received back in our studies the paragraph on that was added in the revised version of the manuscript, please see in lines: 213-217, 219-2020, 242-243, 302-344, 496-498, 628-644, 663-666, 672-680.
I would suggest applying a theoretical framework. There are a lot of models that are connected with meat consumption.
Thank you for the comment. The figures about the results reflect only which meat type and how often it was consumed. These data could have been gained from the national statistical office or Eurostat. Other questions are more important, such as which factors determine meat consumption.
The sample size is very low. Thus, this study is rather qualitative. At this low sample size, it is not allowed to apply percentages.
Thank you for the valuable comment. Based on the Reviewer comment the paragraph was added in the revised version of the manuscript, please see lines: 197-198, 213-217, 219-220, 302-344, 628-644, 663-666, 672-680.
Reviewer 3 Report
The current study proposes the consumption of animal products that consumption was unsustainable for unprocessed and processed meat, and also for fish. On the contrary, egg consumption was revealed as the most sustainable. However, there are some issues which need to be addressed as suggested below for a more complete research. Please check the attached file.

Author Response
Review 3
( ) I would not like to sign my review report
(x) I would like to sign my review report
English language and style
( ) Extensive editing of English language and style required
( ) Moderate English changes required
(x) English language and style are fine/minor spell check required
( ) I don't feel qualified to judge about the English language and style
|
Yes |
Can be improved |
Must be improved |
Not applicable |
|
|
Does the introduction provide sufficient background and include all relevant references? |
(x) |
( ) |
( ) |
( ) |
|
Are all the cited references relevant to the research? |
(x) |
( ) |
( ) |
( ) |
|
Is the research design appropriate? |
(x) |
( ) |
( ) |
( ) |
|
Are the methods adequately described? |
(x) |
( ) |
( ) |
( ) |
|
Are the results clearly presented? |
(x) |
( ) |
( ) |
( ) |
|
Are the conclusions supported by the results? |
(x) |
( ) |
( ) |
( ) |
Comments and Suggestions for Authors
The current study proposes the consumption of animal products that consumption was unsustainable for unprocessed and processed meat, and also for fish. On the contrary, egg consumption was revealed as the most sustainable. However, there are some issues which need to be addressed as suggested below for a more complete research. Please check the attached file.
Submission Date
23 August 2022
Date of this review
21 Sep 2022 13:59:19
We would like to thank the Reviewer for the constructive and valuable comments. The detailed responses are placed in the relevant places below.
The current study proposes the consumption of animal products that consumption was unsustainable for unprocessed and processed meat, and also for fish. On the contrary, egg consumption was revealed as the most sustainable. However, there are some issues which need to be addressed as suggested below for a more complete research. Please check the attached file.
- This study explains the results of consumption of animal products in Poland in the basis of socio-demographic variables. But it needs to emphasize the practical implications and the contributions of this research through the results of this study in detail.
Thank you for the valuable comments comment. Please see the revised version of the manuscript in lines: 119-124, and 315-332.
- Furthermore, since the currently written content only explains the results of statistics, it is necessary to explain with examples how the results of this study can be used in practice in addition to the results. In other words, it needs to practical contribution from the results of this paper and provides a discussion of how these findings can be applied to relative practitioners.
Thank you for the valuable comment. Please see the revised version of the manuscript in lines 333-344.
- Since this data is from 2017, it will need to be supplemented by updating the data through recent data, statistics, news, or magazines. Although it was a paper written well in effort, it is necessary to explain empirical contributions and to explain the validity of this research in detail.
Thank you for your valuable comment. Please see the revised version of the manuscript in lines 283-290.
Round 2
Reviewer 2 Report
The paper is improved notably, but the research is not enough proper. The paper deals with only a sample of 67 respondents. There is a lack of mathematical-statistical methods in the analysis. Hypotheses must be numbered and verified at the end of the paper. The scientific value can be increased by at least 40 more respondents and applying e.g. cluster analysis or at least Khi-square statistics.
I suggest further reading: Besarta Vladi ; Elena Kokthi ; Gert Guri ; Anikó Kelemen-ErdÅ‘s: Mapping Stakeholders Perceptions on Innovation Skills, through the Borich Needs Assessment Model: Empirical Evidence from a Developing Country, ACTA POLYTECHNICA HUNGARICA (1785-8860 1785-8860): 19 8 pp 49-68 (2022)
Author Response
Open Review 2.2.
(x) I would not like to sign my review report
( ) I would like to sign my review report
English language and style
( ) Extensive editing of English language and style required
(x) Moderate English changes required
( ) English language and style are fine/minor spell check required
( ) I don't feel qualified to judge about the English language and style
|
Yes |
Can be improved |
Must be improved |
Not applicable |
|
|
Does the introduction provide sufficient background and include all relevant references? |
(x) |
( ) |
( ) |
( ) |
|
Are all the cited references relevant to the research? |
(x) |
( ) |
( ) |
( ) |
|
Is the research design appropriate? |
( ) |
( ) |
(x) |
( ) |
|
Are the methods adequately described? |
( ) |
(x) |
( ) |
( ) |
|
Are the results clearly presented? |
( ) |
(x) |
( ) |
( ) |
|
Are the conclusions supported by the results? |
( ) |
(x) |
( ) |
( ) |
Comments and Suggestions for Authors
The paper is improved notably, but the research is not enough proper. The paper deals with only a sample of 67 respondents. There is a lack of mathematical-statistical methods in the analysis. Hypotheses must be numbered and verified at the end of the paper. The scientific value can be increased by at least 40 more respondents and applying e.g. cluster analysis or at least Khi-square statistics.
I suggest further reading: Besarta Vladi; Elena Kokthi; Gert Guri; Anikó Kelemen-ErdÅ‘s: Mapping Stakeholders Perceptions on Innovation Skills, through the Borich Needs Assessment Model: Empirical Evidence from a Developing Country, ACTA POLYTECHNICA HUNGARICA (1785-8860 1785-8860): 19 8 pp 49-68 (2022)
Submission Date
23 August 2022
Date of this review
29 Sep 2022 15:43:09
We would like to thank the Reviewer for the constructive and valuable comments. The detailed responses to the Reviewer’s comments are placed in the relevant places below.
The paper is improved notably, but the research is not enough proper. The paper deals with only a sample of 67 respondents. There is a lack of mathematical-statistical methods in the analysis. Hypotheses must be numbered and verified at the end of the paper. The scientific value can be increased by at least 40 more respondents and applying e.g. cluster analysis or at least Khi-square statistics.
I suggest further reading: Besarta Vladi; Elena Kokthi; Gert Guri; Anikó Kelemen-ErdÅ‘s: Mapping Stakeholders Perceptions on Innovation Skills, through the Borich Needs Assessment Model: Empirical Evidence from a Developing Country, ACTA POLYTECHNICA HUNGARICA (1785-8860 1785-8860): 19 8 pp 49-68 (2022)
Thank you for the valuable comments. The suggested article has been read. Regarding the statistical analyses. In our questionnaire surveys, most of the questions requested multiple answers. Therefore, statistical analyses (like Chi-test) weren't possible due to the variation of the total n number (as respondents chose multiple answers) in different questions. As only the part of the survey related to the marking of eggs chosen by respondents required choosing only one answer the Chi-squared test was performed for defining the relation between the marking of eggs bought and the socio-demographic characteristics of the respondents. The results of the Chi-squared test were added in the manuscript in Tables 2-5. Also related paragraphs were added in lines 119-124, 195-211 and 214-222, 309-315, 544-547, 616-618, 681-690, 693-697 (tracked changes doc file). Please see the revised version of the manuscript.
Reviewer 3 Report
Modifications have been made as requested. I agree to publish the paper in the journal.
Author Response
Open Review 3.2.
( ) I would not like to sign my review report
(x) I would like to sign my review report
English language and style
( ) Extensive editing of English language and style required
( ) Moderate English changes required
(x) English language and style are fine/minor spell check required
( ) I don't feel qualified to judge about the English language and style
|
Yes |
Can be improved |
Must be improved |
Not applicable |
|
|
Does the introduction provide sufficient background and include all relevant references? |
(x) |
( ) |
( ) |
( ) |
|
Are all the cited references relevant to the research? |
(x) |
( ) |
( ) |
( ) |
|
Is the research design appropriate? |
(x) |
( ) |
( ) |
( ) |
|
Are the methods adequately described? |
(x) |
( ) |
( ) |
( ) |
|
Are the results clearly presented? |
(x) |
( ) |
( ) |
( ) |
|
Are the conclusions supported by the results? |
(x) |
( ) |
( ) |
( ) |
Comments and Suggestions for Authors
Modifications have been made as requested. I agree to publish the paper in the journal.
Submission Date
23 August 2022
Date of this review
04 Oct 2022 03:18:15
We would like to thank the Reviewer for the positive evaluation of the revised version of the manuscript.